# Pectolinarin Inhibits the Bacterial Biofilm Formation and Thereby Reduces Bacterial Pathogenicity

**DOI:** 10.3390/antibiotics11050598

**Published:** 2022-04-29

**Authors:** Daseul Kim, Ki-Young Kim

**Affiliations:** 1Graduate School of Biotechnology, Kyung Hee University, Seocheon, Giheung, Yongin 17104, Korea; charybde@naver.com; 2College of Life Science, Kyung Hee University, Seocheon, Giheung, Yongin 17104, Korea

**Keywords:** *E. faecalis*, *E. faecium*, bacterial biofilm formation, pectolinarin, antibacterial activity

## Abstract

Bacterial biofilms are a growing problem as it is a major cause of nosocomial infection from urinary catheters to chronic tissue infections and provide resistance to a variety of antibiotics and the host’s immune system. The effect of pectolinarin on the biofilm formation in *Enterococcus faecalis*, *Enterococcus faecium*, *Escherichia coli*, *Streptococcus mutans*, *Streptococcus sobrinus*, *Staphylococcus aureus*, *Pseudomonas aeruginosa*, *Cutibacterium acnes*, and *Porphyromonas gingivalis* was studied in TSBg (tryptic soy broth supplemented with 1% glucose). Pectolinarin inhibited biofilm formation of *E. faecalis* (IC_50_ = 0.39 μg/mL), *E. faecium* (IC_50_ = 0.19 μg/mL), *E. coli* (IC_50_ = 0.25 μg/mL), *S. mutans* (IC_50_ = 1.2 μg/mL), *S. sobrinus* (IC_50_ = 1.4 μg/mL), *S. aureus* (IC_50_ = 0.39 μg/mL), *P. aeruginosa* (IC_50_ = 0.9 μg/mL), *P. acnes* (IC_50_ = 12.5 μg/mL), and *P. gingivalis* (IC_50_ = 9.0 μg/mL) without inhibiting the bacterial growth. Pectolinarin also showed increased susceptibility of antibacterial activity with commercially available antibiotics including ampicillin, vancomycin, streptomycin, and oxytetracyclin against *E. faecalis* and *E. faecium*. Finally, pectolinarin dose-dependently reduced the expression of genes including cytolysin genes (*cylLS*, *cylR2* and *cylM*), quorum sensing (QS) genes (*fsrB*, *fsrC*, *gelE*, *ebpA*, *ebpB*, *acm*, *scm* and *bps*), and biofilm virulence genes (*esp*) of *E. faecalis* and *E. faecium*. Pectolinarin reduced the bacterial biofilm formation, activated the antibacterial susceptibility, and reduced the bacterial adherence. These results suggest that bacterial biofilm formation is a good target to develop the antibacterial agents against biofilm-related infections.

## 1. Introduction

Bacterial biofilms are microbial communities encased within a complex matrix and capable of colonizing natural body surfaces such as the epithelium, lungs, and heart, as well as implanted medical devices such as central venous and urinary catheters, intrauterine devices, and prosthetic heart valves [1]. The biofilm offers many advantages to bacteria, including the ability to acquire increased resistance toward antibiotics, which is of particular importance. This resistance leads to complications in the management of biofilm infections and limits therapeutic options [2,3,4]. In addition, bacterial biofilms pose a challenge to the host immune system [5]. During the past few decades, *Enterococcus* spp. have emerged as important healthcare-associated pathogens. The continuing progress of modern medical care with the overuse of antibiotics has undoubtedly contributed to increase the emergence of antibiotic resistance among clinical *Enterococcus* spp. isolates including *E. faecium*, which is essentially more drug-resistant than *E. faecalis*, with more than half of the isolates appearing resistant to the antibiotics [6]. Healthcare-associated infections due to *E. faecalis* and *E. faecium* more frequently showed resistance to high-level vancomycin and ampicillin, and unsusceptibility to antibiotics [7,8]. Therefore, it is necessary to control bacterial biofilm formation to control the bacterial infection [1,8,9].

Pectolinarin is a flavonoid group, which is present in *Cirsium* spp. that is commonly known as plume thistle (Figure 1). *Cirsium* spp. has been reported with bioactive potential including antidiabetic, antioxidant, hepatoprotective, anti-inflammatory, vasorelaxant, and anti-cancer properties [10,11].

In this work, pectolinarin was tested for anti-biofilm formation of pathogenic bacteria for the candidate as an antibiotic’s adjuvant. Pectolinarin showed inhibitory effect of the biofilm formation and thereby increasing the susceptibility of antibiotics. These results indicate that pectolinarin has the potential as an antibacterial adjuvant to treat the biofilm-related infections.

## 2. Results

### 2.1. Pectolinarin Inhibited the Biofilm Formation of Bacteria

The bacterial biofilm is important for bacteria to survive from the treatment of antibiotics or a hard environment. Pectolinarin was tested and showed the dose-dependent inhibition of bacterial biofilm formation by *E. faecalis* (IC_50_ = 0.39 μg/mL), *E. faecium* (IC_50_ = 0.19 μg/mL), *E. coli* (IC_50_ = 0.25 μg/mL), *S. mutans* (IC_50_ = 1.2 μg/mL), *S. sobrinus* (IC_50_ = 1.4 μg/mL), *S. aureus* (IC_50_ = 0.39 μg/mL), *P. aeruginosa* (IC_50_ = 0.9 μg/mL), *P. acnes* (IC_50_ = 12.5 μg/mL), and *P. gingivalis* (IC_50_ = 0.9 μg/mL) (Table 1, Figure 2, Appendix A).

Pectolinarin exhibited the strongest inhibitory effect of biofilm formation against *E. faecalis* and *E. faecium*, so those bacteria were used for further studies.

### 2.2. Pectolinarin Increased the Susceptibility of E. faecalis and E. faecium to Commercialized Antibiotics

Biofilm formation was performed as described above, and the bacterial growth in the presence or absence of pectolinarin (PEC) with ampicillin (AMP), vancomycin (VAN), oxytetracycline (OXY), or streptomycin (STR) was tested after an additional 24 h. Pectolinarin increased the bacterial susceptibility to antibiotics regardless of whether the bacteria showed resistance against the antibiotics (Figure 3). An amount of 1.56 μg/mL of pectolinarin treatment significantly activated the antibacterial activity of ampicillin, which reduced the viable cells to 0.9% compared with 9% of the only-ampicillin treatment condition. Pectolinarin treatment also reduced the bacterial viability by approximately 10% to 20% compared with only oxytetracycline, streptomycin, or vancomycin treatment (Figure 3a). A total of 0.39 μg/mL of pectolinarin treatment also approximately increased by 10% to 20% the antibacterial activity of ampicillin, oxytetracycline, streptomycin, and vancomycin against *E. faecium* (Figure 3b).

### 2.3. Pectolinarin Reduced Bacterial Adherence to T24 Cells in a Dose-Dependent Manner

Pectolinarin showed significant inhibition of biofilm formation at 0.01–50 μg/mL. Microscopic analysis revealed that pectolinarin inhibited adhesion of the bacteria to the human urinary bladder cancer T24 cells (Figure 4) without inhibition of the growth of T24 cells (data not shown). Especially, 0.01 μg/mL of pectolinarin treatment decreased *E. faecium* adherence by 26% compared to planktonic bacteria.

### 2.4. Pectolinarin Did Not Affect Bacterial Growth

Several antimicrobial compounds also inhibit bacterial biofilm formation because antibiotics kill the bacteria and indirectly decrease biofilm formation [12]. To test whether the biofilm formation inhibition of pectolinarin was due to the antibacterial activity, *E. faecalis* and *E. faecium* were treated with 100 μg/mL of pectolinarin for 24 h. Pectolinarin, despite its notable inhibition of biofilm formation, did not show any effect on the bacterial growth (Figure 5).

### 2.5. Pectolinarin Inhibited the Expression of Genes Related to the Biofilm Formation and Virulence of Bacteria

To understand the molecular basis of pectolinarin inhibition of biofilm formation, the expression of genes related to biofilm-associated factors, Cytolysin and QS system, was tested by qRT-PCR. After treatment with 0.01 to 25 μg/mL of pectolinarin, the expressions of biofilm-associated factors including *ebpB* (IC_50_ = 0.09 μg/mL), *esp* (IC_50_ = 0.01 μg/mL), and *gelE* (IC_50_ = 0.01 μg/mL) in *E. faecalis* and *acm* (IC_50_ = 0.01 μg/mL), bps (IC_50_ = 0.01 μg/mL), *ebpA* (IC_50_ = 0.01 μg/mL), *esp* (IC_50_ = 0.09 μg/mL), *gelE* (IC_50_ = 0.09 μg/mL), and *scm* (IC_50_ = 0.01 μg/mL) in *E. faecium* were significantly decreased (Figure 6a and Figure 7a, respectively). The expressions of *fsrB* (IC_50_ = 0.01 and 0.09 μg/mL) and *fsrC* (IC_50_ = 0.01 μg/mL) genes comprising the Fsr quorum-sensing system were reduced by treatment of pectolinarin (Figure 6b and Figure 7b). The treatment of pectolinarin also dose-dependently reduced the expression of genes related with *Enterococcal* cytolysin synthesis including *cylR2* (IC_50_ = 0.09 μg/mL), *cylM* (IC_50_ = 0.09 μg/mL), and *cylLS* (IC_50_ = 0.09 μg/mL) (Figure 6c).

## 3. Discussion

Pathogenic bacteria such as *E. faecalis*, *E. faecium*, *E. coli*, *P. gingivalis*, *S. mutans*, *S. sobrinus*, *S. aureus*, *C. acnes*, and *P. aeruginosa* contribute to the important global cause of nosocomial infections including community- and hospital-acquired infections [7]. This causes community spreading of pathogenic bacteria that leads to a large increase in the at-risk and high cost to controlling pathogens [7]. The bacterial biofilm reduced the antibacterial susceptibility that make it difficult to eradicate pathogenic bacteria. *Enterococcus* spp. in biofilms are more resistant to antibiotics than planktonic *enterococci* are [8,13]. The development of natural products with bioactive ingredients will, therefore, help overcome the issue of drug resistance in bacteria [14]. *Cirsium* species are considered edible plants and are used as various ailments including hemorrhaging, jaundice, and gastrointestinal disorders [10,11]. Pectolinarin is a secondary metabolite of *Cirsium* spp. with demonstrated biological activities including antimicrobial, antioxidant, antidiabetic, and anti-inflammatory activity [10,11]. In this study, pectolinarin was identified as an inhibitor of biofilm formation caused by bacteria including *E. faecalis*, *E. faecium*, *E. coli*, *P. gingivalis*, *S. mutans*, *S. sobrinus*, *S. aureus*, *C. acnes*, and *P. aeruginosa* (Figure 2). Pectolinarin also showed increasing susceptibility of an antibacterial effect in combination with antibiotics. Even though *E. faecalis* and *E. faecium* showed resistance against vancomycin, pectolinarin decreased their growth by approximately 20%. Pectolinarin dose-dependently decreased bacterial adherence to T24 cells (Figure 4). These results suggest that pectolinarin should be useful as the inhibitor of infection of *E. faecalis* and *E. faecium*. Pectolinarin dose-dependently inhibited the QS system and virulence factor expression (Figure 6 and Figure 7). Virulence factors contribute to the pathogenesis of *E. faecalis* and *E. faecium* and many of them were involved in bacterial adhesion to host cells or abiotic surfaces, leading to biofilm formation. In addition, genes for biofilm-associated factors and virulence factors including *acm*, *scm*, *ebpA*, *ebpB*, *esp*, *bps*, *gelE*, *cylR2*, *cylLS*, and *cylM* were significantly down-regulated by the treatment of pectolinarin (Figure 6 and Figure 7). Therefore, the QS system, virulence factor expression, bacterial adherence, and antibiotic resistance were connected with each other, and thus, biofilm inhibition is for not only the biofilm inhibition but also the reduction in the bacterial survival and infection.

In conclusion, pectolinarin inhibited the bacterial QS system (Figure 6 and Figure 7) to block biofilm formation (Figure 2), enhanced the antibiotic susceptibility (Figure 3), and reduced bacterial adherence to the human host cell (Figure 4). Additional studies including the determination of the molecular mechanism, the relation between adherence, biofilm formation, and virulence, and preclinical animal experiments increase the proof for pectolinarin as an agent for a new antibacterial agent or adjuvant.

## 4. Materials and Methods

### 4.1. Bacterial Strains

Strains used in this study are listed in Table 1. *E. faecalis*, *E. faecium*, *E. coli*, and *S. aureus* were maintained in tryptic soy broth (TSB) and tryptic soy broth supplemented with 1% glucose (TSBg). *S. mutans* and *S. sobrinus* were maintained in Brain Heart Infusion (BHI) broth. *P. gingivalis* was maintained in tryptic soy broth supplemented with 10% defibrinated horse blood agar. *P. aeruginosa* was maintained in nutrient agar, and *C. acnes* was maintained in Reinforced Clostridial agar. All cultures were incubated at 37 °C [13,15,16].

### 4.2. Biofilm Formation Assay

Biofilm formation was performed using TSBg as previously described [12,15,17,18,19,20,21]. *E. faecalis*, *E. faecium*, *E. coli*, *S. mutans*, *S. sobrinus*, *S. aureus*, *P. aeruginosa*, *C. acnes*, and *P. gingivalis* were added (OD_600_ = 0.1) to individual wells of 96-well flat-bottomed plates. Pectolinarin with concentrations ranging from 0.01 to 100 μg/mL was added to respective wells, and the cells were incubated at 37 °C for 24 h. The inhibition of biofilm formation by compound was detected by the crystal violet staining method. Briefly, after 24 h of treatment, the supernatant was removed and the wells were rinsed with physiological saline. In addition, 1% crystal violet (CV) solution was added to each well and incubated for 30 min. The excess of dye was removed by washing the plates under running water. The bound CV was released by adding 125 μL of 30% acetic acid followed by an incubation for 15 min at room temperature. The absorbance was measured at 595 nm using a microplate reader (Bio Tek Instruments, CA, USA) [15,22].

### 4.3. The Combinatorial Antibacterial Effects of Pectolinarin with Commercialized Antibiotics

The antibacterial effects by the combinatorial treatment of pectolinarin with antibiotics were evaluated using the plate-counting method. The bacterial culture at an absorbance of OD_600_ = 0.1 was prepared, and 1 mL of the bacterial suspension with pectolinarin (final concentrations of 1.56, 0.39, 0.09, and 0.01 μg/mL) was added. Planktonic bacteria were removed using sterile saline and the medium (TSBg and Pectolinarin) was incubated for 24 h. The bacteria in the biofilm were enumerated using the plate-counting method [12].

### 4.4. Bacterial Adherence Assays

The human urinary bladder carcinoma T24 cell was purchased from the Korean Cell Line Bank (Seoul, Korea). T24 cells were maintained in RPMI 1640 media supplemented with 10% fatal bovine serum in a 5% CO_2_ incubator at 37 °C [12,23,24]. T24 cells were added to 24-well plates (0.5 × 10^6^ cells per well) for 24 h prior and co-incubated with 0.01–50 µg/mL of pectolinarin-treated *E. faecium* or *E. faecalis* (100 MOI, 5 × 10^7^ CFU per well) for 24 h. Cells were then washed three times with PBS. Gram iodine mordant was applied for 1 min and briefly washed with PBS. To remove any nonspecific crystal violet, a Gram decolorizer solvent was added to the plate for 30 s. After washing with PBS until the PBS ran clear, safranin was applied for 30 min. Samples were assessed under a microscope [25,26].

### 4.5. Time Kill Assay

Growth curves were obtained as previously described with slight modification [22]. *E. faecalis* and *E. faecium* cultures were prepared with tryptic soy broth at an OD_600_ = 0.1. Pectolinarin (100 μg/mL) was added and then the cells were incubated at 37 °C. Growth was evaluated by measuring the optical density of OD_600_ using a microplate reader after 0, 1, 2, 4, 8, 12, 24, and 72 h [11].

### 4.6. Quantitative RT-PCR Analysis

Bacterial cultures and the pectolinarin treatment condition was the same as the method to check the biofilm formation inhibition. Total RNA was isolated using TRIzol reagent (Life Technology, Thermo Fisher Scientific, MA, USA) according to the manufacturer’s instructions, and the reverse transcriptase (NanoHelix, Daejeon, Korea) reaction was prepared using 1 µg of RNA to obtain cDNA. qRT-PCR was carried out using the 2X Sybr Green qPCR Mater Mix (CellSafe, Yongin, Korea). Primer sets for the *fsrB*, *fsrC*, *gelE*, *acm*, *scm*, *ebpA*, *ebpB*, *esp*, *bps*, *cylM*, *cylR2*, and *cylLS* genes are listed in Table 2. Genes encoding 23 sRNA and *tufA* were used as endogenous controls.

### 4.7. Statistical Analysis

All experiments were performed at least three times, and all data are represented as the mean ± S.D.

## 5. Conclusions

Pectolinarin showed the reduced bacterial biofilm formation, activated the susceptibility of commercially available antibacterial agents, and reduced the bacterial adherence to host cells. These results recommend the bacterial biofilm formation as a good target to develop the antibacterial agents against biofilm-related infections with low cytotoxicity, and pectolinarin should be a promising candidate.

## Figures and Tables

**Figure 1 antibiotics-11-00598-f001:**
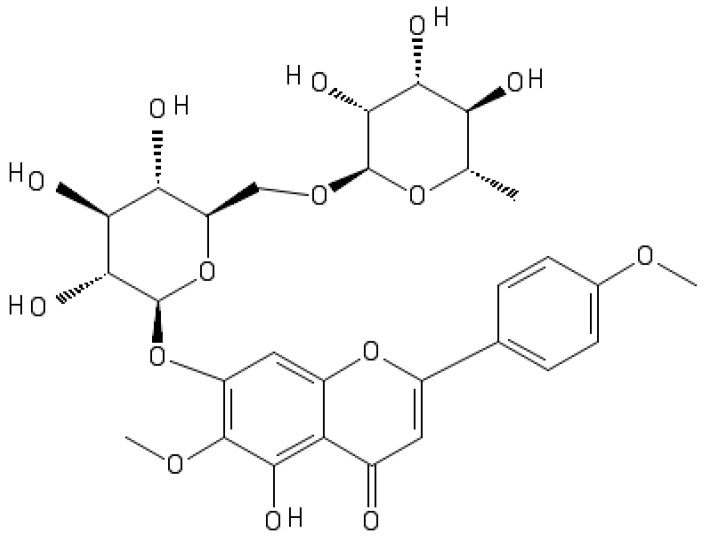
Chemical structure of pectolinarin.

**Figure 2 antibiotics-11-00598-f002:**
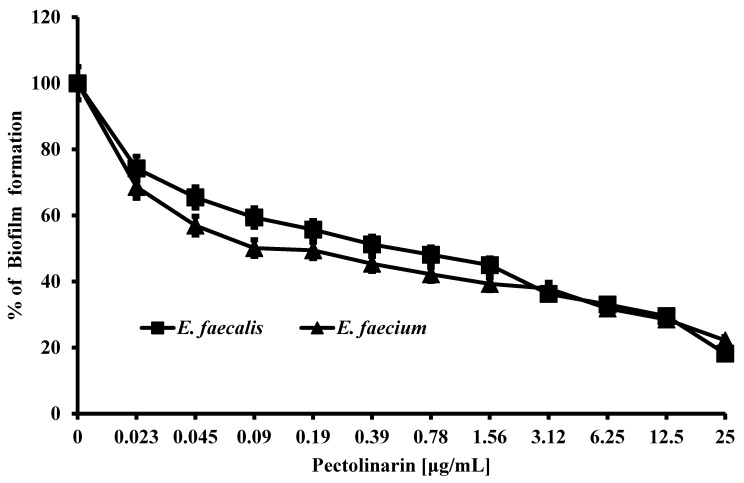
Pectolinarin inhibited biofilm formation of *E. faecalis* and *E. faecium.* Biofilm of *E. faecalis* and *E. faecium* was induced in medium supplemented with pectolinarin at the indicated concentrations at 37 °C for 24 h. Data are the mean values ± SD of triplicate experiments.

**Figure 3 antibiotics-11-00598-f003:**
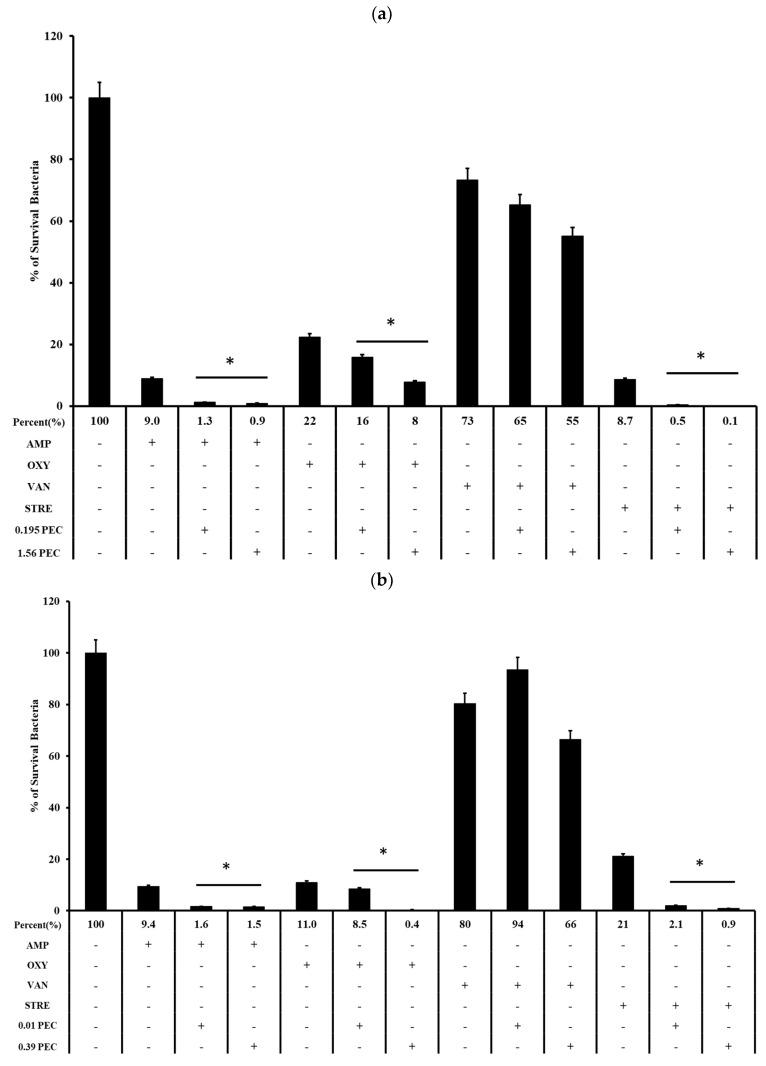
Pectolinarin increased antibiotic susceptibility. (**a**) Pectolinarin increased the susceptibility of *E. faecalis* to conventional antibiotics. Biofilms formed for 24 h by growing *E. faecalis* in TSBg followed by treatment with OXY (3.125 μg/mL), AMP (6.25 μg/mL), STR (100 μg/mL), or VAN (100 μg/mL) alone or in combination with pectolinarin (0.39 and 1.56 μg/mL) for 24 h. (**b**) Pectolinarin increased the susceptibility of *E. faecium* to conventional antibiotics. Biofilms formed for 24 h by growing *E. faecium* in TSBg followed by treatment with OXY (3.125 μg/mL), AMP (6.25 μg/mL), STR (100 μg/mL), or VAN (100 μg/mL) alone or in combination with pectolinarin (0.01 and 0.39 μg/mL) for 24 h. Data are the mean values ± SD of triplicate experiments. * = *p* < 0.05 was considered significant.

**Figure 4 antibiotics-11-00598-f004:**
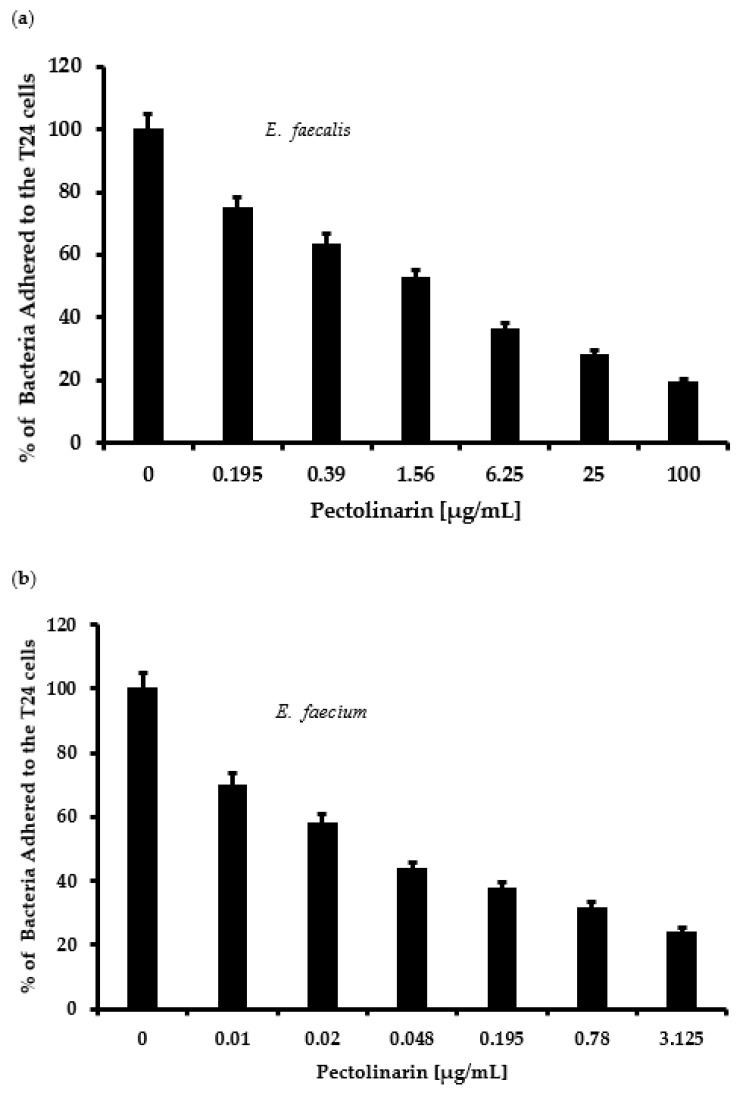
Pectolinarin reduced bacterial adherence to host cells. (**a**) Pectolinarin reduced *E. faecalis* adherence to the host cells. After *E. faecalis* was allowed to infect T24 cells for 24 h, cultures were washed with PBS, and adhered bacterial cells were counted by plating. (**b**) Pectolinarin reduced *E. faecium* adherence to the host cells. After *E. faecium* was allowed to infect T24 cells for 24 h, cultures were washed with PBS, and adhered bacterial cells were counted by plating. Data are the mean values ± SD of triplicate experiments.

**Figure 5 antibiotics-11-00598-f005:**
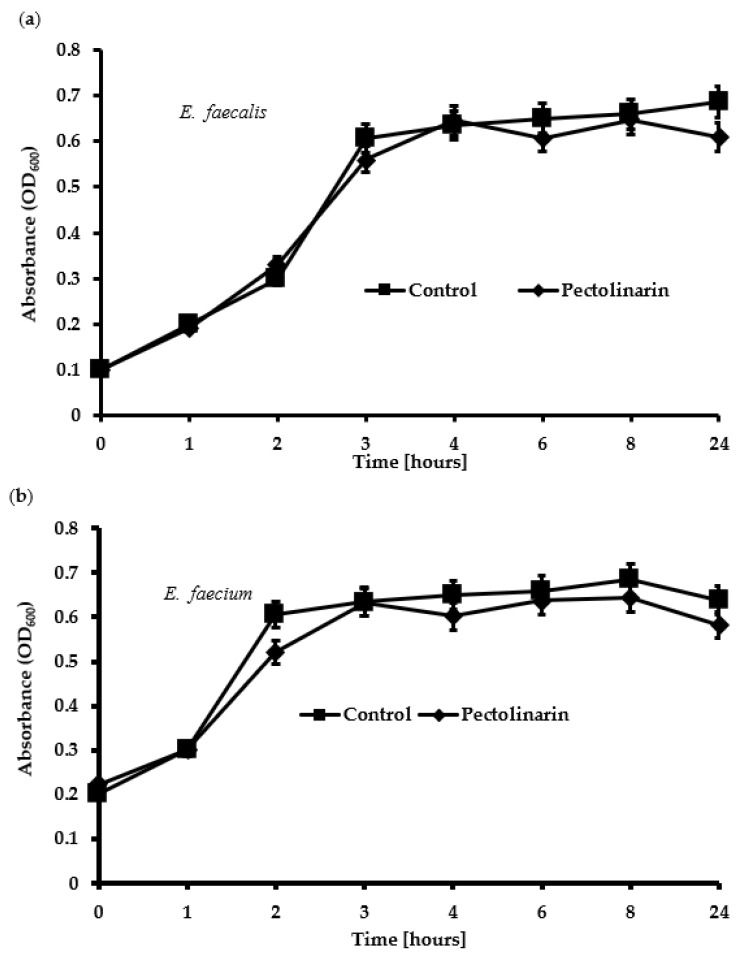
Pectolinarin did not inhibit the growth of *E. faecalis* and *E. faecium*. (**a**) 100 μg/mL of pectolinarin with OD_600_ = 0.1 of *E. faecalis* was incubated at 37 °C for 24 h. (**b**) 100 μg/mL of pectolinarin with OD_600_ = 0.1 of *E. faecium* was incubated at 37 °C for 24 h. The means and standard deviations from at least triplicated determinations are represented. Data are the mean values ± SD of triplicate experiments.

**Figure 6 antibiotics-11-00598-f006:**
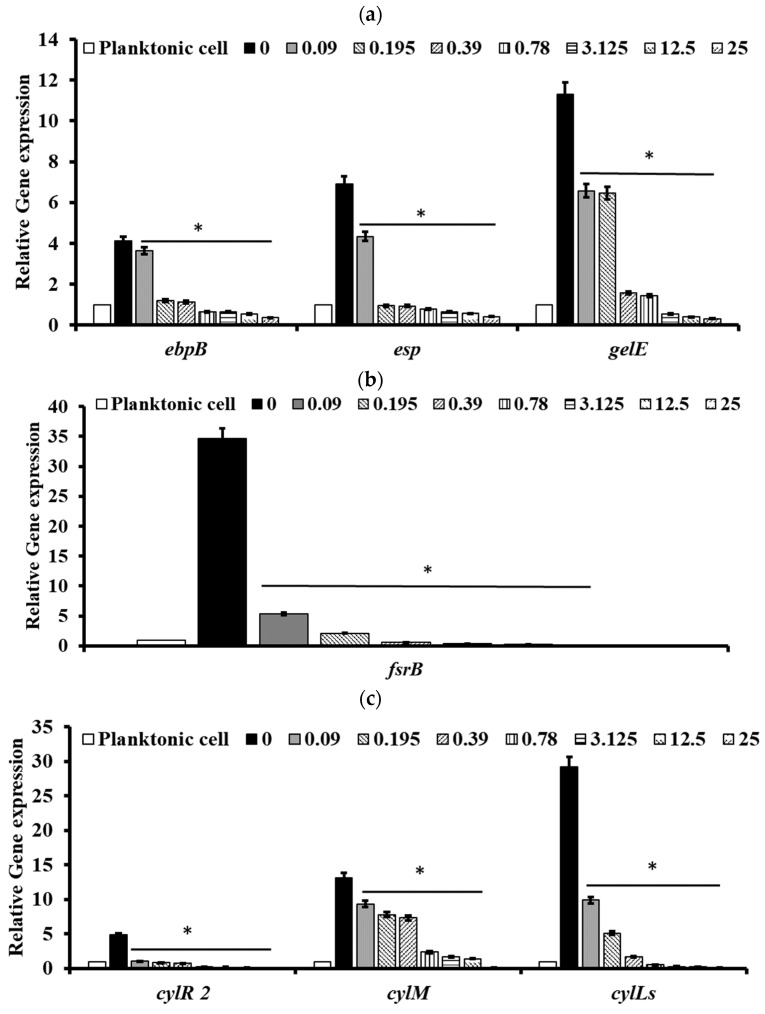
Pectolinarin inhibited the expression of genes related to the biofilm formation and bacterial virulence in *E. faecalis*. (**a**) Pectolinarin inhibited the expression of genes for biofilm formation in *E. faecalis*. (**b**) Pectolinarin inhibited the expression of genes for the quorum-sensing system in *E. faecalis*. (**c**) Pectolinarin inhibited the expression of genes for the bacterial virulence in *E. faecalis*. Total RNA was extracted from biofilm control bacteria and the biofilm-induced bacteria that were treated with the indicated concentration of pectolinarin, converted to cDNA, and analyzed by qPCR with the respective primers. The means and standard deviations from at least triplicate determinations are represented. * = *p* < 0.05 was considered significant.

**Figure 7 antibiotics-11-00598-f007:**
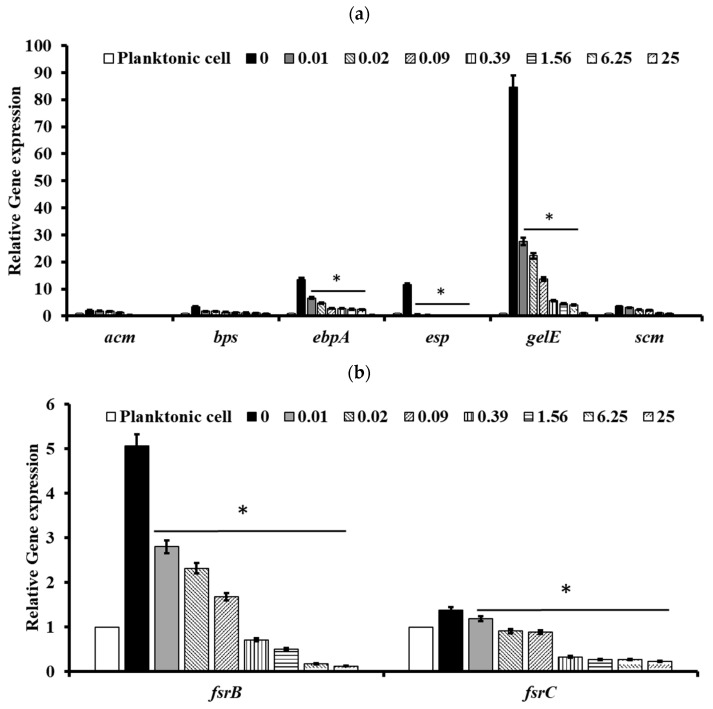
Pectolinarin inhibited the expression of genes related to the biofilm formation and the bacterial virulence in *E. faecium*. (**a**) Pectolinarin inhibited the expression of genes for biofilm formation in *E. faecium*. (**b**) Pectolinarin inhibited the expression of genes for the quorum-sensing system in *E. faecium*. Total RNA was extracted from biofilm control bacteria and the biofilm-induced bacteria that were treated with the indicated concentration of pectolinarin, converted to cDNA, and analyzed by qPCR with the respective primers. The means and standard deviations from at least triplicate determinations are represented. * = *p* < 0.05 was considered significant.

**Table 1 antibiotics-11-00598-t001:** Pentoliniums inhibited biofilm formation of bacteria spp. (IC_50_ value).

Strain	Strain Number	IC_50_ (Half Maximal Inhibitory Concentration; μg/mL)	Source
*Enterococcus faecalis*	CCARM 5511	0.39	Purchased from KACC (Korean Agricultural Culture Collection, Wanju, Korea), CCARM (Culture Collection of Antimicrobial Resistant Microbes, Seoul, Korea), or KCTC (Korean Collection for Type Cultures, Daejon, Korea)
*Enterococcus faecium*	KACC11954	0.19	
*Escherihia coli*	KACC11598	0.25	
*Streptococcus mutans*	KACC16833	1.2	
*Streptococcus sobrinus*	CCARM3506	1.4	
*Staphylococcus aureus*	KCTC5809	0.39	
*Pseudomonas aeruginosa*	KACC14021	0.9	
*Cutibacterium acnes*	CCARM9009	12.5	
*Porphyromonas gingivalis*	KCTC5352	0.9	

**Table 2 antibiotics-11-00598-t002:** Gene-specific primers used for real-time RT-PCR.

Genes	Primer Sequence: 5′ to 3′	Function	Reference
For *E. faecium*			
*esp*	F: CCACGAGTTAGAGGGAACAG R: TTGGAGCCCCATCTTTTTCA	Biofilm formation	[13]
*bps*	F: TATCAGCAACAAGCGGTCAA R: AATCCTGCCCTTTTTCGATT	Biofilm formation	[27]
*fsrC*	F: GCTTATTTGGAAGAACAACGTATCAA R:CGAAACATCGCTAGCTCTTCGT	Efae regulator	[12]
*gelE*	F: CGGAACATACTGCCGGTTTAGA R: TGGATTAGATGCCACCCGAAAT	Gelatinase	[12]
*fsrB*	F: TGCTCAAAAAGCAAAGCCTTATAA R: GATGACGAGACCGTAGAGTATTACTGAA	Efae regulator	[12]
*ebpA*	F: ACCAAGCCAGACGAAATAGAAGAAG R: ATTGTTTTGGTCAGGTGCATCATAGA	Biofilm-associated pili	[27]
*acm*	F: TCAGCAGTAATGTCACTTCGTTG R: GAATAGGCTGTTCATCTGCTCG	Gelatinase	[28]
*scm*	F: CTAACTGGTAACTATGGCTTGT R: GTCCGTGCTGTCACTTGT	Gelatinase	[28]
*tufA*	F: TACACGCCACTACGCTCAC R: AGCTCCGTCCATTTGAGCAG	Housekeeping gene	[12]
For *E. faecalis*			
*gelE*	F: CGFAACATACTCAACGTTTGAC R: TGGATTAGATGCADDDGAAAT	Gelatinase	[21]
*esp*	F: GCATCAGTATTAGTTGGT R: TTCCTTGTAACACATCAC	Biofilm formation	[21]
*fsrB*	F: TGCYCAAAAAGCAAAGCCTTATAA R: GATGACGAGACCGTAGAGTATTACTGAA	Efae regulator	[21]
*ebpB*	F: CGTACAGGAGGCAAGTCTTT R: AGGTATTCCCCGCTTGATTT	Biofilm-associated pili	[21]
*cylLS*	F: CTGTTGCGGCGACAGCT R: CCACCAACCCAGCCACAA	Cytolysin toxin	[21]
*cylR2*	F: TTTATTTTTATTGGATATCATTCTGTAGTC R: TTCGCTCATCTTTTTTGAATACAG	Cytolysin regulatory	[21]

## Data Availability

The data presented in this study are available on request from the corresponding author.

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
