# Peer review of "Pectolinarin Inhibits the Bacterial Biofilm Formation and Thereby Reduces Bacterial Pathogenicity"

_antibiotics, 2022, doi:10.3390/antibiotics11050598_

Round 1
Reviewer 1 Report
This paper is not particularly innovative because it constitutes a follow-up study of a plant metabolite that has already been reported to inhibit biofilm of other bacteria, against enterococci. However, it does provide new information on the mode of action.
Several other bioactivities have also been reported for the compound in the literature, indicating that this is not an ideal candidate for biofilm inhibition. I would in particular like to learn the MIC of the compound against the tested organisms in a regular serial dilution assay because this can give hints on selectivity. An ideal biofilm inhibitor for combination therapy should actually not have any direct killing effect on the bacteria to avoid generation of resistance.
The experiments performed as such are rather sound. The paper could thus be considered for publication, once the numerous typos, grammar and syntax errors have been corrected. I mad some annotation on the pdf but this is by far not exhaustive.

Author Response
Pectolinarin inhibits the bacterial biofilm formation, thereby increases the antibacterial agent susceptibility, and reduces the bacterial pathogenicity.
We thank the editors and the reviewers for their thoughtful and helpful comments. We have addressed, in a point-by-point manner, all the suggestions and queries from the journal and the reviewer and marked with red in manuscript. The input from the reviewers has allowed us to improve the clarity and quality of our paper. We have included below our point-by-point response to the reviewers’ comments and have included these additions and alterations to the revised manuscript.
Comments and Suggestions for Authors
- Delete these 1) - 4).
: Thanks for your suggestion. We changed the part you mentioned and corrected it.
Bacterial biofilms are a growing problem as it is a major cause of nosocomial infection from urinary catheters to chronic tissue infections and provide resistance to a variety of antibiotics and the host's immune system. The effect of pectolinarin on the biofilm formation in Enterococcus faecalis, Enterococcus faecium, Escherichia coli, Streptococcus mutans, Streptococcus sobrinus, Staphylococcus aureus, Pseudomonas aeruginosa, Cutibacterium acnes and Porphyromonas gingivalis was studied in TSBg (tryptic soy broth supplemented with 1% glucose). Pectolinarin inhibited biofilm formation of E. faecalis (IC50 =0.39μg/ml), E. faecium (IC50 =0.19μg/ml), E. coli (IC50 =0.25μg/ml), S. mutans (IC50 =1.2μg/ml), S. sobrinus (IC50 =1.4μg/ml), S. aureus (IC50 =0.39μg/ml), P. aeruginosa (IC50 =0.9μg/ml), P. acnes (IC50 =12.5μg/ml), and P. gingivalis (IC50 =9.0μg/ml) without inhibiting the bacterial growth. Pectolinarin also showed synergistic antibacterial activity with commercially available antibiotics including ampicillin, vancomycin, streptomycin, and oxytetracyclin against E. faecalis and E. faecium. Finally, pectolinarin dose-dependently reduced the expression of genes including cytolysin genes (cylLS, cylR2, cylM), quorum sensing (QS) genes (fsrB, fsrC gelE, ebpA, ebpB, acm, scm, bps), and biofilm virulence genes (esp) of E. faecalis and E. faecium. Pectolinarin reduced the bacterial biofilm formation, activated the antibacterial susceptibility, and reduced the bacterial adherence. These results suggest that bacterial biofilm formation is a good target to develop the antibacterial agents against biofilm-related infections.
- due to protective reservoir ??? please rephrase!
: Thank you for your careful work. As you mentioned, we changed.
Bacterial biofilms are a growing problem as it is a major cause of nosocomial infection from urinary catheters to chronic tissue infections and provide resistance to a variety of antibiotics and the host's immune system.
- You should state "the effects of p. on the biofilm formation in ... was studied and explain TSBg in the abstract when using it for the first time.
: Thank you for your constructive suggestion. We changed the part you mentioned.
The effect of pectolinarin on the biofilm formation in Enterococcus faecalis, Enterococcus faecium, Escherichia coli, Streptococcus mutans, Streptococcus sobrinus, Staphylococcus aureus, Pseudomonas aeruginosa, Cutibacterium acnes and Porphyromonas gingivalis was studied in TSBg (tryptic soy broth supplemented with 1% glucose).
- Spell out the genus names at least when using them for the first time! You should also not use the same letter for abbreviation of different genera
: Thank you for your professional advice. We modified the word as you mentioned.
The effect of pectolinarin on the biofilm formation in Enterococcus faecalis, Enterococcus faecium, Escherichia coli, Streptococcus mutans, Streptococcus sobrinus, Staphylococcus aureus, Pseudomonas aeruginosa, Cutibacterium acnes and Porphyromonas gingivalis was studied in TSBg (tryptic soy broth supplemented with 1% glucose).
- “Bacteria“ “Bacterial”
: Thanks for your suggestion. We modified the word as you mentioned.
- faecalis, E. faecium, Bacterial biofilm formation, Pectolinarin, Antibacterial activity
- I would not start a sentence with an abbreviation. Spell it out here!
: Thank you for your careful work. As you mentioned, we changed.
The continuing progress of modern medical care toward more intensive and invasive medical therapies has undoubtedly contributed to increase antibiotic resistance among clinical enterococci isolates including E. faecium which is essentially more drug-resistant than E. faecalis, with more than half of the isolates appearing resistance to the antibiotics [6].
- This does not make sense. Please reprase!
: Thank you for your careful work. As you mentioned, we changed
In this work, pectolinarin was tested for anti-biofilm formation of pathogenic bacteria for the candidate as an antibiotic’s adjuvant.
- Either always in capitals or (preferable) non capitalized
: Thanks for your suggestion.
Pectolinarin in here is first letter but if pectolinrin is not first letter, we changed to a lower case.
Pectolinarin showed inhibitory effect of the biofilm formation and thereby increasing the susceptibility of antibiotics.
- “antibiotic susceptibility” “susceptibility of antibiotics”
: Thank you for your professional advice. As you mentioned, we changed.
Pectolinarin showed inhibitory effect of the biofilm formation and thereby increasing the susceptibility of antibiotics.
- “were” “was”
: Thank you for your careful work. As you mentioned, we changed the word.
Pectolinarin was tested and showed the dose-dependent inhibition of bacterial biofilm formation by Enterococcus faecalis (IC50 =0.39μg/ml), Enterococcus faecium (IC50 =0.19μg/ml), Escherichia coli (IC50 =0.25μg/ml), Streptococcus mutans (IC50 =1.2μg/ml), Streptococcus sobrinus (IC50 =1.4μg/ml), Staphylococcus aureus (IC50 =0.39μg/ml), Pseudomonas aeruginosa (IC50 =0.9μg/ml), P. acnes (IC50 =12.5μg/ml), and P. gingivalis (IC50 =0.9μg/ml) (Table 1, Figure 2).
- “typo!”
: Thanks for your suggestion. We changed with abbreviation because your comment, this is not a first appearance.
Pectolinarin was tested and showed the dose-dependent inhibition of bacterial biofilm formation by E. faecalis (IC50 =0.39μg/ml), E. faecium (IC50 =0.19μg/ml), E. coli (IC50 =0.25μg/ml), S. mutans (IC50 =1.2μg/ml), S. sobrinus (IC50 =1.4μg/ml), S. aureus (IC50 =0.39μg/ml), P. aeruginosa (IC50 =0.9μg/ml), P. acnes (IC50 =12.5μg/ml), and P. gingivalis (IC50 =0.9μg/ml)
- “those bacteria used” were
: Thank you for your careful work. We have corrected the part you mentioned.
Pectolinarin exhibited the strongest inhibitory effect of biofilm formation against E. faecalis and E. faecium, so those bacteria were used for further studies.
- did not show ?
: Thank you so much. We’ve attached the data as you mentioned.
Pectolinarin, despite its notable inhibition of biofilm formation, did not show any effect on the bacterial growth (Figure 5).
- add space
: Thank you for your careful work.
Figure 5. Pectolinarin did not inhibit the growth of E. faecalis and E. faecium. (a) 100 μg/mL of pectolinarin with OD600=0.1 of E. faecalis was incubated at 37℃ 24 h. (b) 100 μg/mL of pectolinarin with OD600=0.1 of E. faecium was incubated at 37℃ 24 h.
- “regarding” “related”
: Thank you for your careful work. As you mentioned, we corrected the word.
Pectolinarin inhibited the expression of genes related to the biofilm formation and virulence of bacteria.
- highly than is a no-go
: Thank you so much. We changed the word as you mentioned.
Bacterial biofilm reduced the antibacterial susceptibility to the pathogen that challenge to eradicate from hospital-acquired infections. Enterococcus spp. in biofilms are more resistant to antibiotics than planktonic enterococci [8, 14].
- Cirsium italics
: Thank you for your professional advice. We changed as you mentioned.
Cirsium species are considered edible plants and are used as various ailments including hemorrhaging, jaundice, and gastrointestinal disorders [11, 12].
- “is a Cirsium isolate” “a secondary metaboilte of Cirsium spp.”
: Thank you very much. We changed as you mentioned.
Pectolinarin is a secondary metabolite of Cirsium spp. with demonstrated biological activities including antimicrobial, antioxidant, antidiabetic, and anti-inflammatory activity [11, 12].
- “of” “in combination with”
: Thank you for your professional advice We have added word as you mentioned.
Pectolinarin also showed the synergistic antibacterial effect in combination with antibiotics.
- “suggest” “suggest(or these results)”
: Thank you for your careful work. We have corrected the part you mentioned.
- Table 2 They should all be deposited either at the institute of the authors or in a public collection.
: Thank you for your advice. We used the strain that purchased from KACC, CCARM, and KCTC. Those strains are available for any researcher.
- “curve was” “curves were”
: Thank you very much. We changed.
Growth curves were obtained as previously described with slight modification [23].
- “Bacteria” “Bacterial”
: Thank you for your careful work. We have corrected the part you mentioned.
Bacterial cultures and pectolinarin treatment condition was same with the method to check the biofilm formation inhibition.
- NanoHelix, Korea always also state the city!
: Thank you for your professional advice. we have added a city for the part you mentioned.
NanoHelix, Daejeon, Korea
- Italics, Stains
: Thank you very much. we changed to Italics for the part you mentioned.
Tomita, H.; Ike, Y. Tissue-specific Adherent Enterococcus faecalis stains that show highly efficient adhesion to human bladder carcinoma T24 cells also adhere to extracellular matrix proteins. Infect Immun 2004, Volume 72, 5877-85.
- italics - please check the whole text for formatting errors
: Thank you very much. we changed to Italics for the part you mentioned.
Guenther, F.; Stroh, P.; Wagner, C.; Obst. U.; Hansch. G.M. Phagocytosis of staphylococci biofilms by polymorph nuclear neutrophils: S. aureus and S. epidermidis differ with regard to their susceptibility towards the host defense. Int J Artif Organs 2009, Volume 32, pp. 565-73.
And others also changed to italic style.
Reviewer 2 Report
It would be better if Figure 3 presented the bacterial survival by log survival since this would give a clearer picture of the ability of thePectolinarin to assist the antibiotic in killing the bacteria.
Using percent survival rather than a log of percent survival gives a distorted view of how effective Pectolinarin is in the assistance of bacterial killing. It is much easier to see the effectiveness if presented in a log format. Although I no longer can see the Figure, a good example is the fact that in some cases Pectolinarin appears to cause almost complete killing, while this is nearly 2-3 logs of killing. This leave several logs of bacteria alive.
- What is the main question addressed by the research? The main question addressed was how effective was Pectolinarin in helping known antibiotics to kill bacteria.
- Do you consider the topic original or relevant in the field, and if. so, why? The topic appears to be original since the authors appear to be the ones to first isolate pure pectolinarin
- What does it add to the subject area compared with other published
material? It is a new additive. - What specific improvements could the authors consider regarding the
methodology? This is discussed above for Figure 3 - Are the conclusions consistent with the evidence and arguments
presented and do they address the main question posed? Yes - Are the references appropriate? Yes
- Please include any additional comments on the tables and figures. See above for Figure 3.
Author Response
Pectolinarin inhibits the bacterial biofilm formation, thereby increases the antibacterial agent susceptibility, and reduces the bacterial pathogenicity.
We thank the editors and the reviewers for their thoughtful and helpful comments. We have addressed, in a point-by-point manner, all the suggestions and queries from the journal and the reviewer and marked with red in manuscript. The input from the reviewers has allowed us to improve the clarity and quality of our paper. We have included below our point-by-point response to the reviewers’ comments and have included these additions and alterations to the revised manuscript.
Comments and Suggestions for Authors
Comments and Suggestions for Authors
It would be better if Figure 3 presented the bacterial survival by log survival since this would give a clearer picture of the ability of the Pectolinarin to assist the antibiotic in killing the bacteria.
Using percent survival rather than a log of percent survival gives a distorted view of how effective Pectolinarin is in the assistance of bacterial killing. It is much easier to see the effectiveness if presented in a log format. Although I no longer can see the Figure, a good example is the fact that in some cases Pectolinarin appears to cause almost complete killing, while this is nearly 2-3 logs of killing. This leave several logs of bacteria alive.
- What is the main question addressed by the research?
The main question addressed was how effective Pectolinarin in was helping known antibiotics to kill bacteria.
: The main topic is that Pectolinarin, one of the substances extracted from the plant called Cirsium japonicum var. maackii inhibits the biofilm formation of bacteria, and also influence the bacterial infection.
- Do you consider the topic original or relevant in the field, and if. so, why?
The topic appears to be original since the authors appear to be the ones to first isolate pure pectolinarin
: Bacterial biofilm is the main reason for bacterial infection. If the biofilm formation is blocked, that should cause the reduce virulence and infection. In addition, blocking the biofilm formation should increase the susceptibility of antibiotics. If those are happened, the substance should be work as an adjuvant for the antibiotics with less or no toxicity.
- What does it add to the subject area compared with other published material?
It is a new additive.
: As a growing problem of drug resistance strain, this is not directly killing the pathogenic bacteria. That suggested the using the virulence factor as a new antibiotics or antibiotic adjuvant should not increase the resistance occurrence but reduce the infection with less antibiotics or naturally (Macrophage).
Based on that idea, we are going to gather some more candidate and try to find the combinatorial trial method in mouse model.
- What specific improvements could the authors consider regarding the methodology?
This is discussed above for Figure 3
: We actually compare the biofilm formation assay and now we are trying to compare the method with the ability of resistance because we found that the strain is not a determinant of biofilm formation, rather individual sub-strain showed different amount of biofilm formation ability. Also we provide the co-culture assay for the infection.
- Are the conclusions consistent with the evidence and arguments presented and do they address the main question posed?
Yes
: In the case of the question you mentioned, I think it has been well addressed.
- Are the references appropriate?
Yes
: All of references are appropriate and double checked.
- Please include any additional comments on the tables and figures.
See above for Figure 3.
: We though that the abbreviation and detailed methods are included in manuscript.
Reviewer 3 Report
In this manuscript the authors demonstrate the anti-biofilm activity of Pectolinarin. This study is well thought out but some key controls are missing in vital experiments (for example Figure 4 – the authors failed to determine (or mention) human cell survival post treatment with the antimicrobial and the bacterial cells). Midway through the article the authors stop discussing Pectolinarin and start qRT-PCR against Taxifolin. If the qRT-PCR data is indeed against taxifolin (and not pectolactin i.e., a mistake with the labelling) then this reviewer fails to see the relevance of including such data in this manuscript. If this is the case then the aims and outcomes of the manuscript cannot be realised using the datasets provided and this paper should be rejected. My overall recommendation at this point is major revision until the aforementioned point is clarified. Below I have added specific comments which should be addressed:
Specific Comments:
1) The title should be re-worded as it makes little sense - the latter part of the title seems redundant please consider rephrasing.
2) The manuscript should be written in past tense. Please ammend throughout.
3) Line 11: replace "biofilm is" with "biofilms are".
4) Line 12: urinary or venous catheters, more specificity needed.
5) Line 14: What is TSBg? Please write the abbreviation out fully in the first instance of using it.
6) Line 15: Mention which genes were amplified? and against which bacteria?
7) Line 20: which antibiotics?
8) line 25: remove "the" - bacterial biofilm formation alone is fine.
9) Line 25: The concluding sentence of the abstract should be rewrote. I fail to understand why low cytotoxicity has been mentioned - did the authors test pectolinarin against relevant mammalian cell lines?
10) Line 37: replace enterococcus with "Enterococcus spp.,". Please correct throughout.
11) Line 39: It is the overuse of antibiotics that have led to the emergence of antibiotic resistance, not more invasive medical therapies. Please correct this statement.
12) Line 43: This sentence should be rephrases as I am struggling to understand what it means.
13) Line 45: therefor is spelt incorrectly - it should be "therefore".
14) Line 58: inhibits*
15) Line 67: so those bacteria "were" used for further studies.
16) Table 1: the strain type should be incorporated so the readers know which strain was used (not just the genus and species).
17) Figure 2: were the experiments conducted in triplicates? I see no error bars and therefore no statistical analysis can be conducted?
Further, the authors should consider combining both graphs to make comparisons between the bacteria easier.
18) Line 87: of should be replaced with to.
19) Did the authors calculate the IC50 of the conventional antibiotics? If not why were the concentrations selected used?
20) Figure 3: I am not sure how this figure shows synergistic antibiotic activity - the legend should be rephrased to make this clearer.
21) Figure 4: Did the authors check the viability of the host cells post antimicrobial incubation?
How are the authors sure that the bacterial cells were indeed adhered to the T24 cells and not growing in the human cell growth media?
How many repeats(?) this should be stated in the figure legend.
Which media were the cells plated on?
22) Line 122: restraint should be inhibit.
23) Line 126: any should be replaced with no.
24) Line 128 and throughout: I am unsure as to why the authors switch from pectolinarin to taxifolin midway through the manuscript?
why was taxifolin used to determine its effect on gene experession and not pectolinarin?
25) Figure 5: what is taxifolin?
How many repeats this should be stated in the figure legend.
26) Line 142: How have the authors calculated specific IC50 values for gene expression? I'm unsure this is correct.
27) Figure 6 and 7: is Taxifolin the same as pectolinarin? I am unsure if the authors have accidentally mixed up their figures?
is relative gene expression a percentage? (%)
-"Planktoic" should be replaced with Planktonic.
Statistical analysis should be conducted to look for significant differences?
How many repeats - please state?
28) Line 183: please rephrase to make clearer.
29) Line 199: was this shown as this data (Fig 6+7) was for taxifolin?
30) References: Please ensure the references adhere to the journals guidelines.
Author Response
Pectolinarin inhibits the bacterial biofilm formation, thereby increases the antibacterial agent susceptibility, and reduces the bacterial pathogenicity.
We thank the editors and the reviewers for their thoughtful and helpful comments. We have addressed, in a point-by-point manner, all the suggestions and queries from the journal and the reviewer and marked with red in manuscript. The input from the reviewers has allowed us to improve the clarity and quality of our paper. We have included below our point-by-point response to the reviewers’ comments and have included these additions and alterations to the revised manuscript.
Comments and Suggestions for Authors
Reviewer 3
Comments and Suggestions for Authors
In this manuscript the authors demonstrate the anti-biofilm activity of Pectolinarin. This study is well thought out but some key controls are missing in vital experiments (for example Figure 4 – the authors failed to determine (or mention) human cell survival post treatment with the antimicrobial and the bacterial cells). Midway through the article the authors stop discussing Pectolinarin and start qRT-PCR against Taxifolin. If the qRT-PCR data is indeed against taxifolin (and not pectolactin i.e., a mistake with the labelling) then this reviewer fails to see the relevance of including such data in this manuscript. If this is the case then the aims and outcomes of the manuscript cannot be realised using the datasets provided and this paper should be rejected. My overall recommendation at this point is major revision until the aforementioned point is clarified. Below I have added specific comments which should be addressed:
: We are sorry about the point that reviewer mentioned. Actually, we checked the viability of T24 and THP-1 cells for the toxicity using MTT assay but there is no toxicity. But we missed to mention that in the manuscript. we added that “Microscopic analysis revealed that pectolinarin inhibited adhesion of the bacteria to the human urinary bladder cancer T24 cells (Figure 4) the inhibition of the growth of T24 cells (Data not shown).
We found some more candidate (including taxifolin) and we are going to check the combinatorial effects in mouse model, during that time, I mis-spelled. I fill sorry about that mistake.
1) The title should be re-worded as it makes little sense - the latter part of the title seems redundant please consider rephrasing.
: Thanks for your suggestion. Pectolinarin inhibits the bacterial biofilm formation, and thereby reduces the bacterial pathogenicity.
2) The manuscript should be written in past tense. Please amend throughout.
: Thank you for your constructive suggestion. we edited as you mentioned.
3) Line 11: replace "biofilm is" with "biofilms are".
: Thank you for your professional advice. As you mentioned, we corrected that part.
Bacterial biofilms are a growing problem as it is a major cause of nosocomial infection from urinary catheters to chronic tissue infections and provide resistance to a variety of antibiotics and the host's immune system.
4) Line 12: urinary or venous catheters, more specificity needed.
: Thank you for your careful work. As you mentioned, we corrected that part.
Bacterial biofilms are a growing problem as it is a major cause of nosocomial infection from urinary catheters to chronic tissue infections and provide resistance to a variety of antibiotics and the host's immune system.
5) Line 14: What is TSBg? Please write the abbreviation out fully in the first instance of using it.
: Thank you for your valuable suggestions. We edited the wording as you mentioned.
The effect of pectolinarin on the biofilm formation in Enterococcu faecalis, Enterococcu faecium, Escherichia coli, Streptococcus mutans, Streptococcus sobrinus, Staphylococcus aureus, Pseudomonas aeruginosa, Cutibacterium acnes and Porphyromonas gingivalis was studied in TSBg (tryptic soy broth supplemented with 1% glucose).
6) Line 15: Mention which genes were amplified? and against which bacteria?
: Thank you for your constructive suggestion.
Initially, we separate the purpose, method and results section in abstract but one of reviewer suggested that separation is not necessary in this journal. So based on that we remove and rephrased the abstract, so individual gene name listed at line
7) Line 20: which antibiotics?
: Thank you for your careful work. We edited the wording as you mentioned.
Pectolinarin also showed synergistic antibacterial activity with commercially available antibiotics including ampicillin, vancomycin, streptomycin, and oxytetracyclin against E. faecalis and E. faecium.
8) line 25: remove "the" - bacterial biofilm formation alone is fine.
: Thank you so much. We have corrected the wording as you advised.
Pectolinarin inhibited biofilm formation of E. faecalis (IC50 =0.39μg/ml), E. faecium (IC50 =0.19μg/ml), E. coli (IC50 =0.25μg/ml), S. mutans (IC50 =1.2μg/ml), S. sobrinus (IC50 =1.4μg/ml), S. aureus (IC50 =0.39μg/ml), P. aeruginosa (IC50 =0.9μg/ml), P. acnes (IC50 =12.5μg/ml), and P. gingivalis (IC50 =9.0μg/ml) without inhibiting the bacterial growth.
9) Line 25: The concluding sentence of the abstract should be rewrote. I fail to understand why low cytotoxicity has been mentioned - did the authors test pectolinarin against relevant mammalian cell lines?
: Thank you very much
We tested viability of T23 and THP-1 cells using MTT assay and found no reduction of the growth of cells, but as reviewer suggested and we did not include in manuscript we remove the that.
These results suggest that bacterial biofilm formation is a good target to develop the antibacterial agents against biofilm-related infections.
10) Line 37: replace enterococcus with "Enterococcus spp.,". Please correct throughout.
: Thank you for your valuable suggestions. We edited the wording as you mentioned
During the past few decades, Enterococcus spp. have emerged as important healthcare-associated pathogens. The continuing progress of modern medical care toward more intensive and invasive medical therapies has undoubtedly contributed to increase antibiotic resistance among clinical Enterococcus spp. isolates including E. faecium which is essentially more drug-resistant than E. faecalis, with more than half of the isolates appearing resistance to the antibiotics [6].
11) Line 39: It is the overuse of antibiotics that have led to the emergence of antibiotic resistance, not more invasive medical therapies. Please correct this statement.
: Thank you for your professional advice. We edited the wording as you mentioned
The continuing progress of modern medical care with overuse of antibiotics has undoubtedly contributed to increase the emergence of antibiotic resistance among clinical Enterococcus spp. isolates including E. faecium which is essentially more drug-resistant than E. faecalis, with more than half of the isolates appearing resistance to the antibiotics [6].
12) Line 43: This sentence should be rephrases as I am struggling to understand what it means.
: Thank you for your careful work. We have corrected the part you mentioned.
Healthcare-associated infections due to E. faecalis and E. faecium more frequently showed resistant to high-level vancomycin and ampicillin, and unsusceptible to antibiotics [7, 8].
13) Line 45: therefor is spelt incorrectly - it should be "therefore".
: Thank you for your valuable suggestions. We edited the wording as you mentioned
Therefore, it is necessary to control bacterial biofilm formation to control the bacterial infection [8-10].
14) Line 58: inhibits*
: Thanks for your suggestion.
But as you mentioned, we think the past sentence is more reasonable. So changed
Pectolinarin inhibited the biofilm formation of bacteria
15) Line 67: so those bacteria "were" used for further studies.
: Thank you for your advice. As you mentioned, we changed.
Pectolinarin exhibited the strongest inhibitory effect of biofilm formation against E. faecalis and E. faecium, so those bacteria were used for further studies.
16) Table 1: the strain type should be incorporated so the readers know which strain was used (not just the genus and species).
: Thanks for your suggestion. We changed the word for the part you mentioned and corrected it.
Table 1. Pectolinarin inhibited biofilm formation of bacteria spp. (IC50 value).
|
Strain |
Strain number |
IC50 (Half Maximal Inhibitory Concentration; g/mL) |
Source |
|
Enterococcus faecalis |
CCARM 5511 |
0.39 |
Purchased from KACC (Korean Agricultural Culture Collection), CCARM (Culture Collection of Antimicrobial Resistant Microbes), or KCTC (Korean Collection for Type Cultures) |
|
Enterococcus faecium |
KACC11954 |
0.19 |
|
|
Escherihia coli |
KACC11598 |
0.25 |
|
|
Streptococcus mutans |
KACC16833 |
1.2 |
|
|
Streptococcus sobrinus |
CCARM3506 |
1.4 |
|
|
Staphylococcus aureus |
KCTC5809 |
0.39 |
|
|
Pseudomonas aeruginosa |
KACC14021 |
0.9 |
|
|
Cutibacterium acnes |
CCARM9009 |
12.5 |
|
|
Porphyromonas gingivalis |
KCTC5352 |
0.9 |
17) Figure 2: were the experiments conducted in triplicates? I see no error bars and therefore no statistical analysis can be conducted?
: Thank you for your constructive suggestion. In the case of Figure 2, the experiment was performed three times, but missed to increase the thickness.
Further, the authors should consider combining both graphs to make comparisons between the bacteria easier.
: Thank you for your constructive suggestion. As you mentioned, the graph has been modified to make comparisons between bacteria easier. Please confirm.
18) Line 87: of should be replaced with to.
: Thank you so much. we changed the word as you mentioned.
1.56ug/ml of pectolinarin treatment significantly activated the antibacterial activity of ampicillin, which reduced the viable cells of 0.9% compared with 9% of the only ampicillin treatment condition. Pectolinarin treatment also reduced approximately 10% to 20% to the bacteria viability compare with only oxytetracycline, streptomycin, or vancomycin treatment (Figure 3 a).
19) Did the authors calculate the IC50 of the conventional antibiotics? If not why were the concentrations selected used?
: Thank you for your professional advice. In the case of the concentration used in our experiment, after test the broth microdilution assay to determine the MIC value, use the MIC50 value was used.
20) Figure 3: I am not sure how this figure shows synergistic antibiotic activity - the legend should be rephrased to make this clearer.
: Thank you so much. We edited as you advised.
As you mentioned, the results that we made showed increased susceptibility, but not sure whether that is synergistic or additive effects. So, we remove the synergistic in the manuscript.
21) Figure 4: Did the authors check the viability of the host cells post antimicrobial incubation?
: Thank you for your constructive suggestion. It was tested whether there was cytotoxicity against T24cell and THP-1 with pectolinarin, but we did not see any reduced viability.
How are the authors sure that the bacterial cells were indeed adhered to the T24 cells and not growing in the human cell growth media?
: Thank you so much. We edited as you advised.
After culturing the bacteria using a medium that induces biofilm formation, the T24 cell line was infected with the bacteria, and after three times washing Gram stain was performed to confirm it and observed through a microscope. We did not concerned whether the bacteria grow at the human cell growth media, we just check the bacteria cells attached (or not removed by washing) to human cells.
How many repeats(?) this should be stated in the figure legend.
: Thank you for your valuable suggestions. we edited as you mentioned. All experiments were repeated 3 times as stated in “Statistical analysis”
Which media were the cells plated on?
: Thank you so much. We edited as you advised.
Cells were plated on tryptic soy broth (TSB) plates.
22) Line 122: restraint should be inhibit.
: Thanks for your suggestion. We changed you mentioned.
Several antimicrobial compounds also inhibit bacterial biofilm formation because antibiotics kill the bacteria and indirectly decrease biofilm formation [13].
23) Line 126: any should be replaced with no.
: Thank you for your constructive suggestion. We modified the word as you mentioned.
Pectolinarin, despite its notable inhibition of biofilm formation, did not show any effect on the bacterial growth (Figure 5).
24) Line 128 and throughout: I am unsure as to why the authors switch from pectolinarin to taxifolin midway through the manuscript? why was taxifolin used to determine its effect on gene experession and not pectolinarin?
: Thank you for your constructive suggestion.
We made a mistake while editing my thesis. All data written here is from Pectolinarin.
25) Figure 5: what is taxifolin?
: Thank you for your constructive suggestion. We modified the word as you mentioned.
Figure 5. Pectolinarin did not inhibit the growth of E. faecalis and E. faecium. (a) 100 μg/mL of pectolinarin with OD600=0.1 of E. faecalis was incubated at 37 ℃ 24hours. (b) 100 μg/mL of taxifolin with OD600=0.1 of E. faecium was incubated at 37℃ 24hours. The means and standard deviations from at least triplicated determinations are represented.
How many repeats this should be stated in the figure legend.
: Thank you for your valuable suggestions. we edited as you mentioned. All experiments were repeated 3 times.
26) Line 142: How have the authors calculated specific IC50 values for gene expression? I'm unsure this is correct.
: Thanks for your suggestion.
The IC50 value obtained through in the case of the concentration used in our experiment, after testing the gene expression change, based on how much of gene expression is reduced by treatment of pectolinarin, we calculated how much of the concentration is reach the half of reduction of gene expression.
27) Figure 6 and 7: is Taxifolin the same as pectolinarin? I am unsure if the authors have accidentally mixed up their figures?
: Thank you for your constructive suggestion. We modified the word as you mentioned.
Figure 6. Pectolinarin inhibited the expression of genes related to the biofilm formation and bacterial virulence in E. faecalis.
(a) Pectolinarin inhibited the expression of genes for biofilm formation in E. faecalis. (b) Pectolinarin inhibited the expression of genes for the quorum sensing system in E. faecalis. (c) Pectolinarin inhibited the expression of genes for the bacterial virulence in E. faecalis. Total RNA was extracted from biofilm control bacteria and the biofilm induced bacteria that treated with the indicated concentration of pectolinarin, converted to cDNA, and analyzed by qPCR with the respective primers. The means and standard deviations from at least triplicate determinations are represented.
Figure 7. Pectolinarin inhibited the expression of genes related to the biofilm formation and the bacterial virulence in E. faecium.(a) pectolinarin inhibited the expression of genes for biofilm formation in E. faecium. (b) pectolinarin inhibited the expression of genes for the quorum sensing system in E. faecium. Total RNA was extracted from biofilm control bacteria and the biofilm induced bacteria that treated with the indicated concentration of taxifolin, converted to cDNA, and analysed by qPCR with the respective primers. The means and standard deviations from at least triplicate determinations are represented.
is relative gene expression a percentage? (%)
: Thank you for your valuable suggestions.
The gene expression is compared with the untreated planktonic control as a 1 and so if a sample showed 4, suggested that approximately 4-fold of gene expression is increased.
-"Planktoic" should be replaced with Planktonic.
: Thank you for your constructive suggestion. We edited as you mentioned.
(a)
(b)
Figure 7. Pectolinarin inhibited the expression of genes related to the biofilm formation and the bacterial virulence in E. faecium.(a) Pectolinarin inhibited the expression of genes for biofilm formation in E. faecium. (b) Pectolinarin inhibited the expression of genes for the quorum sensing system in E. faecium. Total RNA was extracted from biofilm control bacteria and the biofilm induced bacteria that treated with the indicated concentration of pectolinarin, converted to cDNA, and analysed by qPCR with the respective primers. The means and standard deviations from at least triplicate determinations are represented.
Statistical analysis should be conducted to look for significant differences?
: Thank you for your valuable suggestions. we edited as you mentioned. All experiments were repeated 3 times.
How many repeats - please state?
: Thank you for your valuable suggestions. we edited as you mentioned. All experiments were repeated 3 times.
28) Line 183: please rephrase to make clearer.
: Thank you so much. We changed.
Bacterial biofilm reduced the antibacterial susceptibility that make difficult to eradicate pathogenic bacteria.
29) Line 199: was this shown as this data (Fig 6+7) was for taxifolin?
: Thanks for your suggestion. We made a mistake while editing my thesis. All data written here is from Pectolinarin.
30) References: Please ensure the references adhere to the journals guidelines.
: Thank you for your constructive suggestion. We modified the References as you mentioned.
- Donlan, R.M.; Costerton, J.W. Biofilm: survival mechanisms of clinically relevant microorganisms. Microbiol. Rev. 2002, 15, 167-193.
- Davies, D. Understanding biofilm resistance to antibacterial agents. Rev. Drug. Discov. 2003, 2, 114-122.
- T.F.; O’Toole, G.A. Mechanisms of biofilm resistance to antimicrobial agents. Trends Microbiol. 2001, 9, 34-39.
- Tomita, H.; Ike, Y. Tissue-specific Adherent Enterococcus faecalis stains that show highly efficient adhesion to human bladder carcinoma T24 cells also adhere to extracellular matrix proteins. Immun. 2004, 72, 5877-5885.
- Guenther, F.; Stroh, P.; Wagner, C.; Obst. U.; Hansch. G.M. Phagocytosis of staphylococci biofilms by polymorph nuclear neutrophils: aureus and S. epidermidis differ with regard to their susceptibility towards the host defense. Int. J. Artif. Organs 2009, 32, 565-573.
- Strickertsson, J.A.B.; Desler, C.; Bertelsen, J.M.; Machado, A.M.D.; Wadstrom, T.; Winther, O.; Rasmussen, L.J.; Hansen, L.F. Enterococcus faecalis infection causes inflammation, intracellular oxphos-independent ROS production, and DNA damage in human gastric cancer cells. PLoS ONE 2013, 8, e63147.
- Hidron, A.I.; Edwards, J.R.; Patel, J.; Horan, T.C.; Sievert, D.M.; Pollock, D.A.; Fridkin, S.K.; National Healthcare Safety Network Team, Participating National Healthcare Network Facilities NHSN annual update: antimicrobial-resistant pathogens associated with healthcare-associated infection: annual summary of data reported to the National Healthcare Safety Network at the Centers for Disease Control Prevention. Control Hosp. Epidemiol. 2008, 29, 996-1011.
- Arias, C.A.; Murray, B.E. The rise of the Enterococcus: beyond vancomycin resistance. Rev. Microbiol. 2012, 10, 266-278.
- Alejandro, T.A.; Valle, J.; Solano, C.; Arrizubieta, M.J.; Cucarella, C.; Lamata, M.; Amorena, B.; Leiva, J.; Penades, J.R.; Lasa, I. The entrococcal surface protein, Esp, is involved in Enterococcus faecalis biofilm formation. Environ. Microbiol. 2001, 67, 4538-4545.
- Lim, H.; Soo, K.H.; Chang, H.W.; Bae, K.; Kang, S.S.; Kim, H.P. Anti-inflammatory activity of pentolinarigenin and pectolinarin isolated form Cirsium chanroenicum. Pharm. Bull. 2008, 31, 2063-2067.
- Luo, H.; Jing, B.H.; King, S.M.; Chen, Y.C. Inhibition of cell growth and VEGF expression in ovarian cancer cells by flavonoids. Cancer 2008, 60, 800-809.
- Top, J.; Paganelli, F.L.; Zhang, X.; Schaik, W.V.; Leavis, H.L.; Asbroek, M.V.L.; Poll, T.V.D.; Leendertse, M.; Bonten, M.J.M.; Willems, R.J.L. The Enterococcus faecium Enterococcal biofilm regulator, EbrB, regulates the esp operon and is implicated in biofilm formation and intestinal colonization. PLoS ONE 2013, 8, e65224.
- Yan, C.; Tangjuan, L.; Ke, W.; Hou, C.; Cai, S.; Huang, Y.; Du, Z.; Huang, H.; Kong, J.; Chen, Y. Baicalein inhibits Staphylococcus aureus biofilm formation and the quorum sensing system in vitro. PLos ONE 2016, 11, e0153469.
- Sreedhar, R.N.; Kavindra, V.S.; Jouko, S.; Danielle, A.G.; Magnus, H.; Stanley, L.E.; Barbara, E.M. Endocarditis and biofilm-associated pili of Enterococcus faecalis. Clin. Invest. 2006, 116, 2799-2807.
- Ken-ichi, A.M.; Kunitoshi, Y.; Yoshimitsu, M.; Tanaka, Y.; Ogura, T. & Sugimoto, S. Inhibitory effects of Myricetin derivatives on curli-dependent biofilm formation in Escherichia coli. Rep. 2018, 8, 8452.
- Ong, T.H.; Chitra, E.; Ramamurthy, S.; Siddalingam, R.P.; Yen, K.H.; Ambu, S.P.; Davamani, F. Chitosan-propolis nanoparticle formulation demonstrates anti-bacterial activity against Enterococcus faecalis PLoS ONE 2017, 12, e0174888.
- Chien, Y.C. Surface sensing for biofilm formation in Pseudomonas aeruginosa. Microbiol. 2017, 8, 2671.
- Hahnel, S.; Muhlbauer, G.; Hoffmann, J.; Ionescu, A.; Burgers, R.; Rosentritt, M.; Handel, G.; Haberlein, I. Streptococcus mutans and Streptococcus sobrinus biofilm formation and metabolic activity on dental materials. Acta Odontol. Scand. 2012, 70, 114-121.
- He, L.; Wang, H.; Zhang, R.; Li, H. The regulation of Porphyromonas gingivalis biofilm formation by ClpP. Biophys. Res. Commun. 2019, 509, 335-340.
- Holmberg, A.; Lood, R.; Morgelin, M.; Soderquist, B.; Holst, E.; Collin, M.; Christensson, B.; Rasmussen, M. Biofilm formation by Propionibacterium acnes is a characteristic of invasive isolates. Microbiol. Infect. 2009, 15, 787-795.
- Mok, J.Y.; Kang, H.J.; Cho, J.K.; Jeon, I.H.; Kim, H.S.; Park, J.M.; Jeong, S.I.; Shim, J.S.; Jang, S.I. Antioxidative and anti-inflammatory effects of extracts from different organs of Cirsium japonicum ussuriense. Kor. J. Herbology 2011, 26, 39-47.
- Kim, J.G.; Ha, Q.B.T.; Shin, Y.K.; Kim, Y.K. Antifungal activity of magnoflorine against Candida World J. Microbiol. Biotechnol. 2018, 34, 167.
- Stewart, P.S. Mechanisms of antibiotic resistance in bacterial biofilms. J. Med. Microbiol. 2002, 292, 107-113.
- Thomas, V.C.; Thurlow, L.R.; Boyle, D.; Hancock, L. Regulation of autolysis-dependent extracellular DNA release by Enterococcus faecalis extracellular proteases influences biofilm development. Bacteriol. 2008, 190, 5690-5698.
- Becerra, S.C.; Roy, D.C.; Sanchez, C.J.; Christy, R.J.; Burmeister, D.M. An optimized staining technique for the detection of Gram positive and Gram-negative bacteria within tissue. BMC Res. Notes 2016, 9, 216.
- Daw, K.; Baghdayan, A.S.; Awasthi, S.; Shankar, N. Biofilm and planktonic Enterococcus faecalis elicit different responses from host phagocytes in vitro. FEMS Immunol. Med. Microbiol. 2012, 65, 270-282.
- Sillanpää, J.k.; Nallapareddy, S.R.; Singh, K.V.; Prakash, P.; Fothergill, T.; That, H.T.; Murray, B.E. Characterization of the ebpfm pilus-encoding operon of Enterococcus faecium and its role in biofilm formation and virulence in a murine model of urinary tract infection. Virulence. 2010, 1, 236.
- Hendrickx, A.P.A.; Luit-Asbroek, M.V.; Schapendonk, C.M.E.; Wamel W.J.B.; Braat, J.C.; Wijnands, L.M.; Bonten, M.J.M.; Willems, R.J.L. SgrA, a Nidogen-Binding LPXTG Surface Adhesin Implicated in Biofilm Formation, and EcbA, a Collagen Binding MSCRAMM, Are Two Novel Adhesins of Hospital-Acquired Enterococcus faecium. Infect Immun. 2009, 77, 5097
Round 2
Reviewer 3 Report
The authors have addressed all of my prior concerns (well done!), however, I do have two further comments which the authors should rectify before publication:
1) remove "the" from the title - "reduces bacterial pathogenicity" makes more sense.
2) The authors should include statistical analysis for all of their graphs (i.e., t-tests/ANOVAs) - and significance should be denoted as asterisks (*). This will allow the readers to quickly determine if the results are significant (this will also improve the robustness of the data analysis and conclusions reached).
Once the aforementioned comments have been addressed I believe that this manuscript will be ready for publication and make a welcome addition to the Antibiotics journal.
Author Response
Pectolinarin inhibits the bacterial biofilm formation, and thereby reduces the bacterial pathogenicity.
We thank the editors and the reviewers for their thoughtful and helpful comments. We have addressed, in a point-by-point manner, all the suggestions and queries from the journal and the reviewer and marked with red in manuscript. The input from the reviewers has allowed us to improve the clarity and quality of our paper. We have included below our point-by-point response to the reviewers’ comments and have included these additions and alterations to the revised manuscript.
Comments and Suggestions for Authors
The authors have addressed all of my prior concerns (well done!), however, I do have two further comments which the authors should rectify before publication:
1) remove "the" from the title - "reduces bacterial pathogenicity" makes more sense.
: Thanks for your suggestion. Based on your advice, we have changed word.
Pectolinarin inhibits the bacterial biofilm formation, and thereby reduces bacterial pathogenicity.
2) The authors should include statistical analysis for all of their graphs (i.e., t-tests/ANOVAs) - and significance should be denoted as asterisks (*). This will allow the readers to quickly determine if the results are significant (this will also improve the robustness of the data analysis and conclusions reached).
: Thanks for your suggestion. We have modified the experimental method for the problematic part.
Figure 2. Pectolinarin inhibited biofilm formation of E. faecalis and E. faecium. Biofilm of E. faecalis and E. faecium was induced in medium supplemented with pectolinarin at the indicated concentrations at 37℃ for 24 h. Data are the mean values ± SD of triplicate experiments.
(a)
(b)
Figure 3. Pectolinarin increased antibiotic susceptibility. (a) Pectolinarin increased the susceptibility of E. faecalis to conventional antibiotics. Biofilms formed for 24 h by growing E. faecalis in TSBg followed by treatment with OXY (3.125μg/mL), AMP (6.25μg/mL), STR (100μg/mL), or VAN (100μg/mL) alone or in combination with pectolinarin (0.39 and 1.56μg/mL) for 24 h. (b) Pectolinarin increased the susceptibility of E. faecium to conventional antibiotics. Biofilms formed for 24 h by growing E. faecium in TSBg followed by treatment with OXY (3.125μg/mL), AMP (6.25μg/mL), STR (100 μg/mL), or VAN (100 μg/mL) alone or in combination with pectolinarin (0.01 and 0.39μg/mL) for 24 h. Data are the mean values ± SD of triplicate experiments (P < 0.05).
(a)
|
(b)
|
Figure 4. Pectolinarin reduced bacterial adherence to host cells. (a) Pectolinarin reduced E. faecalis adherence to the host cells. After E. faecalis was allowed to infect T24 cells for 24 h, cultures were washed with PBS, and adhered bacterial cells were counted by plating. (b) Pectolinarin reduced E. faecium adherence to the host cells. After E. faecium was allowed to infect T24 cells for 24 h, cultures were washed with PBS, and adhered bacterial cells were counted by plating. Data are the mean values ± SD of triplicate experiments (P < 0.05 ).
(a)
|
(b)
|
Figure 5. Pectolinarin did not inhibit the growth of E. faecalis and E. faecium. (a) 100μg/mL of pectolinarin with OD600=0.1 of E. faecalis was incubated at 37℃ 24 h. (b) 100μg/mL of pectolinarin with OD600=0.1 of E. faecium was incubated at 37℃ 24 h. The means and standard deviations from at least triplicated determinations are represented. Data are the mean values ± SD of triplicate experiments.
(a)
(b)
(c)
Figure 6. Pectolinarin inhibited the expression of genes related to the biofilm formation and bacterial virulence in E. faecalis.
(a) Pectolinarin inhibited the expression of genes for biofilm formation in E. faecalis. (b) Pectolinarin inhibited the expression of genes for the quorum sensing system in E. faecalis. (c) Pectolinarin inhibited the expression of genes for the bacterial virulence in E. faecalis. Total RNA was extracted from biofilm control bacteria and the biofilm induced bacteria that treated with the indicated concentration of pectolinarin, converted to cDNA, and analyzed by qPCR with the respective primers. The means and standard deviations from at least triplicate determinations are represented (P < 0.05).
(a)
(b)
Figure 7. Pectolinarin inhibited the expression of genes related to the biofilm formation and the bacterial virulence in E. faecium.(a) Pectolinarin inhibited the expression of genes for biofilm formation in E. faecium. (b) Pectolinarin inhibited the expression of genes for the quorum sensing system in E. faecium. Total RNA was extracted from biofilm control bacteria and the biofilm induced bacteria that treated with the indicated concentration of pectolinarin, converted to cDNA, and analysed by qPCR with the respective primers. The means and standard deviations from at least triplicate determinations are represented (P < 0.05).
Once the aforementioned comments have been addressed I believe that this manuscript will be ready for publication and make a welcome addition to the Antibiotics journal.